

# Quantum oscillations of magnetization in interaction-driven insulators

Andrew A. Allocca[1*] and Nigel R. Cooper[1,2]

**1** TCM Group, Cavendish Laboratory, University of Cambridge, Cambridge, CB3 0HE, UK
**2** Department of Physics and Astronomy, University of Florence, 50019 Sesto Fiorentino, Italy

⋆ aa2182@cam.ac.uk

## Abstract

In recent years it has become understood that quantum oscillations of the magnetization as a function of magnetic field, long recognized as phenomena intrinsic to metals, can also manifest in insulating systems. Theory has shown that in certain simple band insulators, quantum oscillations can appear with a frequency set by the area traced by the minimum gap in momentum space, and are suppressed for weak fields by an intrinsic "Dingle damping" factor reflecting the size of the bandgap. Here we examine quantum oscillations of the magnetization in excitonic and Kondo insulators, for which interactions play a crucial role. In models of these systems, self-consistent parameters themselves oscillate with changing magnetic field, generating additional contributions to quantum oscillations. In the low-temperature, weak-field regime, we find that the lowest harmonic of quantum oscillations of the magnetization are unaffected, so that the zero-field bandgap can still be extracted by measuring the Dingle damping factor of this harmonic. However, these contributions dominate quantum oscillations of magnetization at all higher harmonics, thereby providing a route to measure this interaction effect.

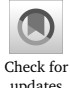

# 1  Introduction

Quantum oscillations (QO) of observables as a function of applied magnetic field have been understood as a phenomenon intimately tied to the idea of a Fermi surface ever since they were first discovered [1]. The well-established Lifshitz-Kosevich theory [2] of QO directly relates the frequency of the oscillations to extremal cross-sectional areas of the Fermi surface. This has allowed the technique to become a useful experimental tool for examining the geometry of Fermi surfaces in materials. However, this long-held understanding has been challenged recently by the measurement of quantum oscillations in insulators, notably the strongly-correlated Kondo insulators $SmB_6$ [3–5] and $YbB_{12}$ [6,7], and the insulating phase of $WTe_2$ [8,9], believed to be an excitonic insulator [10], which all lack a traditional notion of a Fermi surface entirely.

Many theoretical works [11–30] have been put forward in response, seeking to understand the phenomenon in these specific materials and how QO may arise in insulators more generally. In this second direction, a direct calculation shows that generating QO in insulating systems is actually relatively straightforward—if the minimum band gap is not much larger than the cyclotron energy and traces out a nonzero area in the Brillouin zone, then oscillations can be found at the frequency corresponding to this area as though it were a Fermi surface cross section [11,14]. The condition on the ratio of the gap to the cyclotron energy for visibility of these oscillations arises from an intrinsic "Dingle damping" factor–an exponential suppression of the form $\exp(-B_0/B)$, where $B$ is the applied magnetic field strength. In metallic systems the Dingle factor accounts for the effect of disorder; $B_0$ is related to the finite quasiparticle relaxation time [31] and will vary between samples. However, in an insulator $B_0$ is directly related to intrinsic properties the gapped band structure itself. This implies that for band insulators QO contain important information about electronic structure just as they do for metals, and fundamental properties of the band structure may be extracted from careful analysis of the field dependence of oscillation amplitudes.

The question we examine here is whether this result also holds for systems where the band structure is strongly affected by interactions, such as excitonic and Kondo insulators. In the mean field descriptions of these systems at zero field, the mean field parameters obey self-consistent constraints and determine the form of the bands. When a magnetic field is applied these constraints necessarily introduce $B$-dependence to these parameters, which causes the bands themselves to modulate with field and introduces additional contributions to QO not present in 'rigid' band insulators.

To analyze each of these systems we employ the following general procedure. First we analyze the mean field description of the system at zero field, in particular identifying the self-consistent equations that the mean field parameters must obey. With the introduction of a magnetic field we assume that electronic dispersions are quantized into Landau levels and the

mean field parameters acquire $B$-dependent oscillatory components, which for weak fields are small compared to the zero-field parts. We then linearize the self-consistent conditions around the zero-field values and determine the leading effect of the magnetic field on top of the rigid band case.

We find very generally that when considering mean field theories the fundamental harmonic of QO of the magnetization is unaffected, and the oscillatory component of the mean field affects second and higher harmonics only. For both excitonic and Kondo insulator models these new contributions to higher harmonics have exactly the same exponential sensitivity to the size of the gap as for the corresponding rigid band insulators using the fixed $B = 0$ gap, but have different overall dependence on the field strength allowing them to be the dominant contribution to all harmonics to which they contribute.

The remainder of the paper is organized as follows: We begin in Section 2 by examining QO for the case of a rigid band insulator, which is the background around which we linearize in the following sections. In Section 3 we analyze a model excitonic insulator, first applying the mean field approximation at $B = 0$, then considering the oscillations the mean field parameter acquires upon introduction of a magnetic field and it's contributions to QO. In Section 4 we then do the same for the case of a Kondo insulator using the mean field slave-boson formalism.

## 2 Rigid Band Insulator

We first consider a spinless, two-dimensional band insulator at zero temperature described by the Hamiltonian [11]

$$H_0 = \sum_{\mathbf{k}} \Psi_{\mathbf{k}}^{\dagger} \begin{pmatrix} \epsilon_{\mathbf{k}}^c & g \\ g & \epsilon_{\mathbf{k}}^v \end{pmatrix} \Psi_{\mathbf{k}} \,. \tag{1}$$

Here and throughout the rest of our calculations we set $\hbar = 1$, $c$ and $v$ label conduction and valence bands, and $\Psi_{\mathbf{k}} = (\psi_{c,\mathbf{k}}, \psi_{v,\mathbf{k}})^T$, with $\psi_{i,\mathbf{k}}^{\dagger}$ and $\psi_{i,\mathbf{k}}$ the creation and annihilation operators for electrons in band $i$. The conduction band dispersion is $\epsilon_{\mathbf{k}}^c$, which we take to be approximately parabolic in the region of interest, and we set the valence band dispersion to be $\epsilon_{\mathbf{k}}^v = \epsilon_0 - \eta \epsilon_{\mathbf{k}}^c$, with $\eta$ a dimensionless constant and $\epsilon_0$ the shift of the valence band relative to the conduction band. The limit $\eta \to 0$ yields the flat valence band of a heavy fermion system. We take $\epsilon_0 > 0$ so the conduction and valence bands cross and the interband tunneling amplitude $g$ opens a hybridization gap at the band crossing point. The single-particle energies of the system are then

$$E_{\mathbf{k}}^{\pm} = \frac{1}{2} \left( \epsilon_{\mathbf{k}}^c + \epsilon_{\mathbf{k}}^v \pm \sqrt{(\epsilon_{\mathbf{k}}^c - \epsilon_{\mathbf{k}}^v)^2 + 4g^2} \right), \tag{2}$$

which are shown in Fig. 1. We assume a ground state with the lower band entirely filled and the upper band empty, so the system is an insulator. In writing this model we have implicitly assumed that the physics of interest is captured by a two-band model. This is expected as long as any additional bands are well separated in energy from the gap opening point.

Applying a magnetic field $B$ perpendicular to the system quantizes $\epsilon_{\mathbf{k}}^c$ into discrete Landau levels (LL), indexed by $n = 0, 1, 2, \ldots$, via the replacement $\epsilon_{\mathbf{k}}^c \to \epsilon_n^c = (n + \gamma)\omega_c$, with cyclotron frequency $\omega_c$ and phase shift $\gamma$. If $\epsilon_{\mathbf{k}}^c = k^2/2m_c$ exactly, with effective conduction electron mass $m_c$, then this replacement is exact, the cyclotron energy and phase shift are $\omega_c = eB/m_c$ and $\gamma = 1/2$, and $\eta = m_c/m_v$ represents the ratio of effective masses of the two bands. Otherwise this substitution is an approximation valid for weak fields, with $\gamma \in [0, 1)$ in general. Within $E_{\mathbf{k}}^{\pm}$ this replacement gives the energies $E_n^{\pm}$, and sums over momentum are replaced by sums over LL index, $\sum_{\mathbf{k}} \to n_{\Phi} \sum_{n=0}^{\infty}$, with $n_{\Phi} = B/\Phi_0$ the degeneracy of each LL and $\Phi_0 = h/e$ the magnetic flux quantum. Because the hybridization is spatially homogeneous, after these

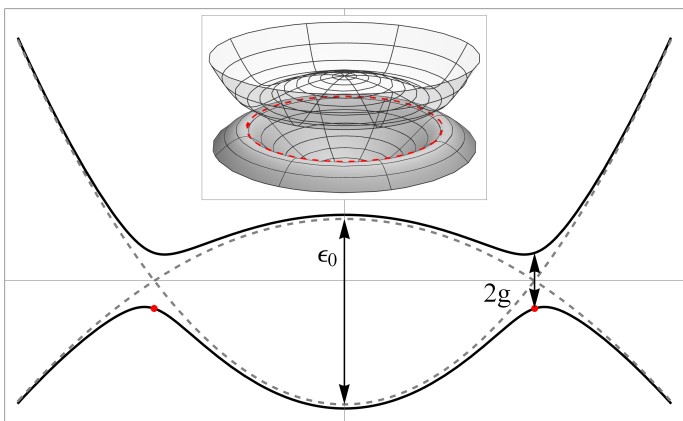

Figure 1: An example band structure of the sort we consider. The solid lines show the energies $E_{\mathbf{k}}^{\pm}$ in Eq. (2), while the dashed lines show $\epsilon_{\mathbf{k}}^{c}$ and $\epsilon_{\mathbf{k}}^{v}$, the two bands in Eq. (1) prior to hybridization. We have indicated the offset energy $\epsilon_0$ and hybridization gap $2g$, and marked in red the point on the lower band where the gap is minimized. Inset: a 3D view of the bands. The area traced by the minimum gap, setting the QO frequency, is indicated with the red dashed line.

replacements the Hamiltonian only couples corresponding Landau levels in the conduction and valence bands, reflected in the form of $E_n^{\pm}$.

At zero temperature the free energy of the system is given by the sum over the energies of all occupied states, which for an insulator is the completely filled lower band. With a magnetic field this energy is

$$\Omega_R(B) = n_\Phi \sum_{n=0}^{\infty} E_n^- , \tag{3}$$

where the subscript $R$ indicates the bands are treated as rigid, with the hybridization unaffected by changing $B$. We can separate $\Omega_R(B)$ into constant $\Omega_{R0}$ and oscillatory $\tilde{\Omega}_R(B)$ parts using the Poisson summation formula [31], which for a general function $f$ is

$$\sum_{n=0}^{\infty} f(n+\gamma) = \int_0^{\infty} dx\, f(x) + 2 \int_0^{\infty} dx \sum_{p=1}^{\infty} f(x) \cos\left(2\pi p(x-\gamma)\right) . \tag{4}$$

Though this system lacks a Fermi momentum and Fermi surface, the momentum which minimizes the band gap characterizes the gapped band structure, and the area in momentum space that it encircles, indicated in Fig. 1, functions in lieu of a Fermi surface for the purposes of QO. We denote this momentum as $k^*$, defined through $d\left(E_{\mathbf{k}}^+ - E_{\mathbf{k}}^-\right)/dk\big|_{k=k^*} = 0$. From this we define a corresponding (noninteger) reference value of the LL index $n^*$ through $\epsilon_{\mathbf{k}^*}^c = \epsilon_{n^*}^c$, giving $n^* = \epsilon_0/\omega^* - \gamma$, where we put $\omega^* = (1+\eta)\omega_c$. For weak magnetic fields we have $n^* \gg 1$, which allows us to find an approximate analytic form of $\tilde{\Omega}_R(B)$,

$$\tilde{\Omega}_R(B) \approx \frac{2|g|n_\Phi}{\pi} \sum_{p=1}^{\infty} \frac{\cos(2\pi p n^*)}{p} K_1\left(2\pi p \frac{2|g|}{\omega^*}\right) \sim \sqrt{\frac{|g|\omega^*}{2}} \frac{n_\Phi}{\pi} \sum_{p=1}^{\infty} \frac{\cos(2\pi p n^*)}{p^{3/2}} e^{-2\pi p \frac{2|g|}{\omega^*}} , \tag{5}$$

where $K_i$ are the modified Bessel functions of the second kind. The final expression here uses the asymptotic form $K_i(x) \sim \sqrt{\pi/2x} \exp(-x)$ for $x \gg 1$, which further refines the weak field regime to the condition $\omega^* \ll 2|g|$, cyclotron energy much smaller than the band gap. This result is very similar in form to the results in Ref. [32] which examines a system with a superconducting gap. It can be derived from the $T \to 0$ limit of the expression in Ref. [11] as

shown in Appendix A. We see that for weak magnetic fields the harmonics of the band insulator free energy are exponentially suppressed by powers of $\exp\left(-2\pi\frac{2|g|}{\omega^*}\right)$, which we identify as the Dingle factor in an insulating system. This factor will function as a small parameter when considering additional oscillatory contributions in the following sections.

## 3 Excitonic Insulator

We now consider the case of an excitonic insulator [33–36]. This type of system is formed from the condensation of excitons with binding energy greater than the inherent band gap of the system, so that the band gap is (predominantly) generated by electron-electron interactions. In the mean field approximation there is a single parameter controlling the insulating properties of the system, the exciton condensate amplitude, which allows for a very simple treatment of QO in the weak field regime.

To describe this type of system we start from a two-band, two-dimensional model Hamiltonian for spinless electrons with an interband interaction,

$$H = \sum_{\mathbf{k}} \Psi_{\mathbf{k}}^{\dagger} \begin{pmatrix} \epsilon_{\mathbf{k}}^c & 0 \\ 0 & \epsilon_{\mathbf{k}}^v \end{pmatrix} \Psi_{\mathbf{k}} - V \sum_{\mathbf{k},\mathbf{k}'} \psi_{c,\mathbf{k}}^{\dagger} \psi_{v,\mathbf{k}} \psi_{v,\mathbf{k}'}^{\dagger} \psi_{c,\mathbf{k}'} \,, \tag{6}$$

where $V$ is the strength of the short-range exciton pairing potential. We decouple the interaction via a mean field approximation neglecting fluctuations, defining the exciton condensate order parameter $\Delta = V \sum_{\mathbf{k}} \left\langle \psi_{v,\mathbf{k}}^{\dagger} \psi_{c,\mathbf{k}} \right\rangle$, where $\langle \cdots \rangle$ denotes the expectation value in the state with a filled lower band and empty upper band. Though generally complex, we can choose $\Delta$ to be purely real and positive by adjusting the phases of $\psi_{c,v}$. We then obtain the excitonic insulator Hamiltonian

$$H_X = \sum_{\mathbf{k}} \Psi_{\mathbf{k}}^{\dagger} \begin{pmatrix} \epsilon_{\mathbf{k}}^c & -\Delta \\ -\Delta & \epsilon_{\mathbf{k}}^v \end{pmatrix} \Psi_{\mathbf{k}} + \frac{\Delta^2}{V} \,, \tag{7}$$

with $\Delta$ obeying the BCS-type gap equation

$$\frac{1}{V} = \sum_{\mathbf{k}} \frac{1}{\sqrt{\left(\epsilon_{\mathbf{k}}^c - \epsilon_{\mathbf{k}}^v\right)^2 + 4\Delta^2}} \,. \tag{8}$$

Note that the fermionic part of Eq. (7) is the same as Eq. (1) with $g = -\Delta$.

This two-dimensional model and our main results can in principle be extended to three dimensions by including additional dispersion along $k_z$, the direction of the magnetic field, as described in e.g. Ref. [31]. Such an extension should not change the fundamental nature of our results. We also note that the role of fluctuations about the mean field order could be considered by extending the mean-field theory (7) via standard means [37–39].

We now consider applying a perpendicular magnetic field $B$, which quantizes the electronic dispersion into Landau levels as discussed in Section 2. Because we assume $\Delta$ to be spatially homogeneous, we still have coupling only between corresponding Landau levels in the two bands. In contrast to the rigid band insulator, however, the value of the gap $\Delta$ acquires magnetic field dependence because of its relationship to the electronic energies through Eq. (8). We put $\Delta \to \Delta(B) = \Delta_0 + \tilde{\Delta}(B)$, where $\Delta_0$ is the constant value of the order parameter at zero field, solving Eq. (8), $\tilde{\Delta}(B)$ is the part of the order parameter that varies with changing field, and we assume that $\left|\tilde{\Delta}(B)\right| \ll \Delta_0$.

### 3.1 Oscillations in the Linearized Theory

The free energy of an excitonic insulator at zero temperature, which we denote $\Omega_X$, is the sum over energies of all states in the lower band, which has the same form as Eq. (3), plus the

energy of the mean field parameter, the second term in Eq. (7). Unlike for the band insulator, the full dependence of $\Omega_X$ on $B$ is partially implicit through $\tilde{\Delta}(B)$. To find the first corrections on top of the band insulator result and we expand around $\Delta = \Delta_0$, keeping terms up to second order in oscillatory quantities, assumed to be small:

$$\Omega_X(B, \Delta) \approx \Omega_{XR} + \frac{\partial \Omega_{XR}}{\partial \Delta_0} \tilde{\Delta} + \frac{1}{2} \frac{\partial^2 \Omega_{XR}}{\partial \Delta_0^2} \tilde{\Delta}^2 = \Omega_{XR0} + \tilde{\Omega}_{XR} + \frac{\partial \tilde{\Omega}_{XR}}{\partial \Delta_0} \tilde{\Delta} + \frac{1}{2} \frac{\partial^2 \Omega_{XR0}}{\partial \Delta_0^2} \tilde{\Delta}^2, \quad (9)$$

where we identify $\Omega_{XR} = \Omega_X(B, \Delta_0)$. The function $\tilde{\Omega}_{XR}(B)$ is the oscillatory part of $\Omega_{XR}$ and has the same form as $\tilde{\Omega}_R(B)$ given in Eq. (5), but with the replacement $g \to -\Delta_0$. The mean field gap $\Delta_0$ is, by definition, the value for which the action is stationary with respect to variation in $\Delta$ (Eq. (8) is equivalent to $\partial \Omega_{XR0}/\partial \Delta_0 = 0$), so the only term remaining at first order in oscillatory quantities is the rigid band contribution, $\tilde{\Omega}_{XR}$. Therefore, the next largest contribution comes at second order, given by the final two terms. This is a general implication of mean field theory, independent of the choice of system or mean field being considered.

We now consider these next largest terms. For both terms we need the form of $\tilde{\Delta}(B)$, which can be evaluated by analyzing the gap equation. For $B \neq 0$ this has the same form as in Eq. (8) but with the replacements noted above, i.e. $\epsilon_{\mathbf{k}}^c \to \epsilon_n^c = (n + \gamma)\omega_c$, $\Delta \to \Delta_0 + \tilde{\Delta}(B)$, etc. We begin by expanding to first order in $\tilde{\Delta}(B)$

$$\frac{1}{V} \approx n_\Phi \sum_{n=0}^{\infty} \frac{1}{\sqrt{\left(\epsilon_n^c - \epsilon_n^v\right)^2 + 4\Delta_0^2}} - n_\Phi \sum_{n=0}^{\infty} \frac{4\Delta_0}{\left(\left(\epsilon_n^c - \epsilon_n^v\right)^2 + 4\Delta_0^2\right)^{3/2}} \tilde{\Delta}(B) \equiv \alpha(B) + \beta(B)\tilde{\Delta}(B). \quad (10)$$

In the second equality we define the two sums as the functions $\alpha(B)$ and $\beta(B)$. We then rewrite these functions in terms of their constant ($\alpha_0$ and $\beta_0$) and oscillatory ($\tilde{\alpha}$ and $\tilde{\beta}$) parts, which can be evaluated with the Poisson summation formula, and keep terms only to first order in oscillations, giving

$$\frac{1}{V} \approx \alpha_0 + \tilde{\alpha}(B) + \beta_0 \tilde{\Delta}(B). \quad (11)$$

Because the left hand side is a constant, we must have

$$\frac{1}{V} = \alpha_0, \quad (12)$$

$$\tilde{\Delta}(B) = -\frac{\tilde{\alpha}(B)}{\beta_0}. \quad (13)$$

Calculating the explicit forms of $\alpha_0$, $\tilde{\alpha}(B)$, and $\beta_0$ (see Appendix B) verifies that Eq. (12) is exactly Eq. (8), the zero-field gap equation determining $\Delta_0$, and gives the explicit form for $\tilde{\Delta}(B)$ via Eq. (13),

$$\tilde{\Delta}(B) = 2\Delta_0 \sum_{p=1}^{\infty} \cos\left(2\pi p n^*\right) K_0 \left(2\pi p \frac{2\Delta_0}{\omega^*}\right) \sim \sqrt{\frac{\Delta_0 \omega^*}{2}} \sum_{p=1}^{\infty} \frac{\cos\left(2\pi p n^*\right)}{\sqrt{p}} e^{-2\pi p \frac{2\Delta_0}{\omega^*}}, \quad (14)$$

where $\omega^* = (1 + \eta)\omega_c$ as before and the second expression is the asymptotic form for weak fields, $\omega^* \ll 2\Delta_0$. As for the oscillatory part of the free energy, we see that the $p^{\text{th}}$ harmonic comes with $p$ powers of the Dingle factor.

With explicit forms for $\tilde{\Delta}(B)$ and $\tilde{\Omega}_{XR}$, the last quantity we need to evaluate is the second derivative of the $B = 0$ free energy $\Omega_{XR0}$ with respect to $\Delta_0$. Using the gap equation to simplify, we find

$$\frac{1}{2} \frac{\partial^2 \Omega_{XR0}}{\partial \Delta_0^2} = 2 \frac{n_\Phi}{\omega^*}. \quad (15)$$

Putting all of the terms together we find the dominant contribution to the free energy at second order in small oscillatory quantities is

$$\frac{\partial \tilde{\Omega}_{XR}}{\partial \Delta_0}\tilde{\Delta} + \frac{1}{2}\frac{\partial^2 \Omega_{XR0}}{\partial \Delta_0^2}\tilde{\Delta}^2 \sim -\frac{\Delta_0 n_{\Phi}}{2}\cos(4\pi n^*)e^{-4\pi\frac{2\Delta_0}{\omega^*}}, \qquad (16)$$

where we have kept only the oscillatory terms at lowest order in the Dingle factor and discarded a term that is smaller by a factor of $\omega^*/2\Delta_0 \ll 1$. We see that this contributes to the second harmonic of QO of the magnetization. Comparing this to the $p = 2$ term of the rigid band contribution, there is a clear difference in the overall field dependence–the prefactor of the mean field term goes as $B$, whereas the rigid band term goes as $B^{3/2}$, so the rigid band term is smaller by a factor of $\frac{1}{2\pi}\sqrt{\frac{\omega^*}{\Delta_0}} \ll 1$ at small fields. Therefore, for weak fields the oscillations of the mean field order parameter provide the dominant contribution to the second harmonic of the free energy.

The contribution from the mean field is likely dominant for all higher harmonics as well. In addition to other terms, such as those acquired by calculating $\tilde{\Delta}$ at higher orders than the linearized framework presented here, we can write down several terms that have a leading $B$ dependence at a lower power than the corresponding term in $\tilde{\Omega}_{XR}$. First, in the term in Eq. (9) proportional to $\tilde{\Delta}^2$, cross terms between the $q$ harmonic of one factor and the $p-q$ harmonic of the other give contributions to the $p^{\text{th}}$ harmonic that also have $p$ powers of the Dingle factor. All such terms have a coefficient that goes as $B$, making them larger than the corresponding term of $\tilde{\Omega}_{XR}$, going as $B^{3/2}$. Additionally, there will be a term contributing to the $p^{\text{th}}$ harmonic of the free energy the form

$$\frac{\partial^{p-1}\tilde{\Omega}_{XR}}{\partial \Delta_0^{p-1}}\tilde{\Delta}^{p-1} \sim B^{2-\frac{p}{2}}e^{-2\pi p\frac{2\Delta_0}{\omega^*}}, \qquad (17)$$

with $\tilde{\Delta}$ given by Eq. (14). We see that this goes as $B^{2-p/2}$, which for small $B$ is larger than $B^{3/2}$ for all $p \geq 2$, and is larger than $B$ for $p > 2$ (it is $B^{1/2}$ for $p = 3$), as shown in Fig. 2. There is no reason why the mean field contributions such as these should exactly cancel for any harmonic $p$ above the first–we have shown this explicitly for $p = 2$–so for weak fields these mean field terms will dominate for all harmonics $p \geq 2$.

We pause to emphasize an important feature of the result that we have found: There is only a single dimensionless parameter, $\omega^*/\Delta_0$, that controls the size of the quantum oscillations (in both the Dingle damping factor and its multiplicative prefactors) arising from the contributions of both the rigid band and self-consistent mean-field parts of the free energy. Rewriting $\omega^*/\Delta_0 = B/B_0$ so that $B_0 = m_c\Delta_0/((1 + \eta)e)$, the value of $B_0$, proportional to the product of the hybridization gap and the cyclotron mass, can be used to characterize individual materials. Indeed, fitting measurements of quantum oscillations to the form of the Dingle factor–as done with metallic systems to extract mean free paths–would here allow for a direct experimental determination of this quantity for a given material. For $B \sim B_0$ the rigid band and mean field contributions to the higher harmonics ($p \geq 2$) are of the same size; below this the mean field part dominates and above this point our approximations begin to break down. Consequently, we see that the oscillations of the gap that we analyze here cannot be ignored whenever they are present–a system with an interaction generated gap is never accurately described by just the corresponding rigid band structure.

We now briefly comment on how our results relate to those in Refs. [29] and [30], which also analyzes oscillations in an excitonic insulator. There are several key differences between what is done there and what we present here, but there is no obvious disagreement. First, Refs. [28] and [29] focus on electronic transport (a response function) via thermally activated electrons and holes, and not the magnetization (a thermodynamic quantity) which is our main

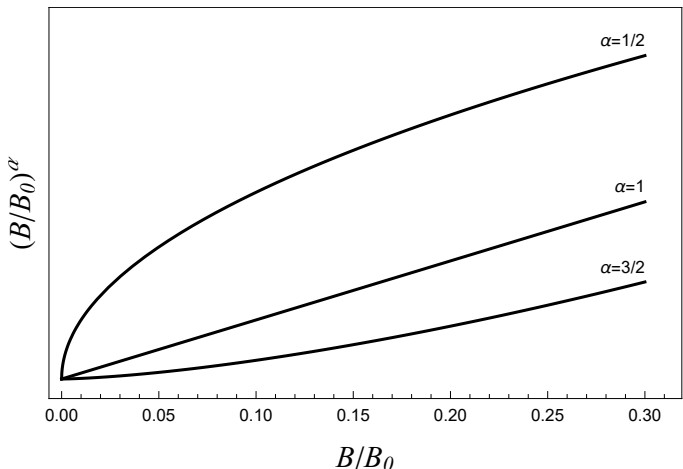

Figure 2: Demonstrating the size of the leading dependencies $B^\alpha$ that we find for various terms, for small $B/B_0 = \omega^*/(2\Delta_0)$. The rigid band case has $\alpha = 3/2$ for all harmonics, while the contributions from the mean field to the second harmonic are larger, with $\alpha = 1$. There are contributions to the third harmonic with $\alpha = 1/2$, which is even larger still.

focus. Second, the calculations there consider a rigid band structure of hybridized particle and hole bands as in Section 2–the fixed hybridization set equal to the excitonic condensate order parameter at $B = 0$–and define the gap for $B \neq 0$ as the energy between the highest energy Landau level in the lower band and the lowest energy Landau level in the upper band. The resulting oscillations of the gap are then more akin to the framework of Ref. [11] than to what we find here.

Because we consider the free energy, and can therefore discuss only thermodynamic quantities, our result cannot be directly compared to those of Refs. [29] and [30]. Importantly, our generic conclusion that there is no interaction contribution to the first harmonic does not apply, and in general one should indeed expect an additional contribution to the fundamental frequency oscillation of non-thermodynamic quantities. How such contributions would compare to the results of Refs. [29] and [30] is left as future work.

## 3.2 Effects of nonzero temperature

All of our calculations so far have been performed for a model at exactly zero temperature. We show in Appendix C that our results apply for a range of nonzero temperatures provided that $T \ll \Delta_0$, well below the transition temperature into the excitonic insulator phase. It is clear that the temperature dependence of quantum oscillations in this model must be quite distinct from the typical Lifshitz-Kosevich form, given by the factor

$$R_T = \frac{p\theta}{\sinh(p\theta)}, \tag{18}$$

where $p$ labels the harmonic and $\theta = 2\pi^2 k_B T/\omega^*$, with $k_B$ Boltzmann's constant. Even for the rigid bandstructure the oscillations show a temperature dependence that departs from the LK form [11]. We expect further corrections beyond this, arising from the effects of self-consistency of the gap $\Delta_0$ as a function of temperature. Computing this dependence would require considering temperatures of the order of the gap, at which point thermal occupation of the upper band would become relevant. This is a regime for which we do not have any analytic results. We note however, that the behaviour is potentially quite rich. The typical expectation

is for temperature to reduce quantum oscillation amplitudes due to thermal broadening of occupied states near the relevant energy. However, here the fact that $\Delta_0$ diminishes with increasing temperature has the potential to counteract that, by lessening the damping from the Dingle factor. We note that a temperature-dependent gap would force one to reconsider the validity of analyzing just the "weak field regime" that we have used so far. For a fixed magnetic field strength and increasing temperature, the ratio $\omega^*/\Delta_0$ grows as $\Delta_0$ diminishes, so for any field strength the regime $\omega^* \sim \Delta_0$ becomes relevant at some temperature. This is certainly a rich avenue for future work, but is beyond the present scope.

## 4 Kondo Insulator

We now look to the case of Kondo insulators, a class of strongly-correlated, heavy fermion system [40] with narrow band gaps first identified over 50 years ago [41]. We begin with an Anderson model in two dimensions [42–45], describing the coupling of a light conduction band to a heavy valence band, localized by strong interactions. The Hamiltonian is

$$
\begin{aligned}
H = &-t \sum_{\langle ij\rangle,\sigma} \left( c_{i\sigma}^{\dagger} c_{j\sigma} + \text{h.c.} \right) - t_d \sum_{\langle ij\rangle,\sigma} \left( d_{i\sigma}^{\dagger} d_{j\sigma} + \text{h.c.} \right) \\
&+ \sum_{i,\sigma} V_i \left( c_{i\sigma}^{\dagger} d_{i\sigma} + d_{i\sigma}^{\dagger} c_{i\sigma} \right) + \sum_i \left( \epsilon_d n_i^d + U n_{i\uparrow}^d n_{i\downarrow}^d \right).
\end{aligned}
\tag{19}
$$

The first line describes the two species of electrons (conduction $c$ and heavy $d$ bands) hopping on a lattice with amplitudes $t$ and $t_d$ respectively, with $t_d < 0$ and $|t_d/t| = \eta \ll 1$. The next term describes interband transitions with amplitude $V_i$, which opens the gap in the spectrum. The final two terms are written in terms of the $d$-electron densities, $n_{i,\sigma}^d = d_{i,\sigma}^{\dagger} d_{i,\sigma}$ and $n_i^d = n_{i,\uparrow}^d + n_{i,\downarrow}^d$. The $\epsilon_d$ term gives the shift of the heavy electron band relative to the conduction band. The $U$ term is a Hubbard interaction between $d$-electrons, forbidding double occupancy in the large $U$ limit. For $U \to \infty$ this condition can be enforced with the slave-boson formalism: put $d_{i\sigma}^{\dagger} = f_{i\sigma}^{\dagger} b_i$, where $f_{i\sigma}$ is a new fermionic degree of freedom, which we refer to as $f$-electrons, and $b_i$ is the slave boson. Each site contains either a boson or a single $f$-electron, and the Hubbard term is replaced by $\sum_i \lambda_i (\sum_\sigma f_{i\sigma}^{\dagger} f_{i\sigma} + b_i^{\dagger} b_i - 1)$, where $\lambda_i$ is a Lagrange multiplier field enforcing the constraint.

We assume that the interband interaction is spatially uniform, $V_i = V$, and now employ the mean field approximation $\lambda_i \to \langle \lambda_i \rangle \equiv \lambda$, $b_i \to \langle b_i \rangle \equiv b$, and $b_i^{\dagger} \to \langle b_i^{\dagger} \rangle \equiv b$. In the continuum limit, which is a valid approximation when considering weak magnetic fields, we obtain the mean field Hamiltonian

$$
H_K = \sum_{\mathbf{k},\sigma} \Psi_{\mathbf{k}\sigma}^{\dagger} \begin{pmatrix} \epsilon_{\mathbf{k}}^c & bV \\ bV & \epsilon_{\mathbf{k}}^f \end{pmatrix} \Psi_{\mathbf{k}\sigma} + \lambda \left( b^2 - 1 \right).
\tag{20}
$$

Here the $f$-band dispersion is $\epsilon_{\mathbf{k}}^f = \epsilon_d + \lambda - \eta b^2 (\epsilon_{\mathbf{k}}^c - 4t)$, and the limit of immobile heavy fermions, i.e. infinite $f$-band mass, corresponds to $\eta \to 0$. We can identify what we called $\epsilon_0$ in the case of the rigid band insulator with $\epsilon_d + \lambda + 4\eta t b^2$ and what we called $g$ with $bV$.

As written here we are considering an even parity coupling between the bands. We could consider an odd parity coupling instead, $\hat{V}(\mathbf{k}) = V \mathbf{d}(\mathbf{k}) \cdot \hat{\sigma}$ with $\mathbf{d}(\mathbf{k}) = -\mathbf{d}(-\mathbf{k})$, which results in nontrivial topological properties [46, 47]. This choice does not affect the nature of the results we present here, however; the zero-field gap appearing in our final results would have a different form reflecting its origins from an odd parity coupling, but the overall form of the expressions in terms of the size of the gap would remain the same. We continue with the simpler case of even parity coupling considered thus far.

For the Kondo insulator there are two self-consistent equations allowing us to determine the two mean field parameters $b$ and $\lambda$, contrasting with the excitonic insulator considered in Section 3 which has only one. The first equation is simply the constraint imposed by the Hubbard interaction, which in the mean field approximation becomes

$$\sum_{\mathbf{k},\sigma} \left\langle f_{\mathbf{k}\sigma}^{\dagger} f_{\mathbf{k}\sigma} \right\rangle \equiv n_0^f = 1 - b^2 \,, \tag{21}$$

where we have defined $n_0^f$, the total $f$-electron density at $B = 0$. The second constraint follows from the equation of motion for the boson field, which in the mean field approximation becomes a demand that the energy be stationary with respect to variation of $b$,

$$\sum_{\mathbf{k},\sigma} \left\langle \frac{V}{2} \left( c_{k\sigma}^{\dagger} f_{k\sigma} + f_{k\sigma}^{\dagger} c_{k\sigma} \right) - \eta b (\epsilon_{\mathbf{k}} - 4t) f_{k\sigma}^{\dagger} f_{k\sigma} \right\rangle = C_0 + \eta b \left( K_0^f + 4t\, n_0^f \right) = -\lambda b \,. \tag{22}$$

Here we have defined two additional functions,

$$C_0 \equiv \frac{V}{2} \sum_{\mathbf{k},\sigma} \left\langle c_{k\sigma}^{\dagger} f_{k\sigma} + f_{k\sigma}^{\dagger} c_{k\sigma} \right\rangle \,, \tag{23}$$

$$K_0^f \equiv -\sum_{\mathbf{k},\sigma} \epsilon_{\mathbf{k}}^c \left\langle f_{k\sigma}^{\dagger} f_{k\sigma} \right\rangle \,, \tag{24}$$

$C_0$ the interband correlation energy and $\eta K_0^f$ the kinetic energy of the $f$-electrons.

We now consider applying a perpendicular magnetic field $B$ to the system. As discussed in Section 2, this can be done by replacing energies with their Landau quantized versions, and sums over momentum with sums over LL index. We note here specifically that for a generic anisotropic Kondo system the hybridization gap does not necessarily open at a fixed energy unless the $f$-band is completely immobile, $\eta = 0$. Therefore, the conclusions we arrive at only generically apply for $\eta \neq 0$ in the case of an isotropic system.

We assume that the effect of a nonzero field on the mean field parameters is to induce an oscillatory component for each above the value determined at $B = 0$, as we did for the case of the excitonic insulator. Explicitly, we put

$$b \to b(B) = b_0 + \tilde{b}(B), \quad \lambda \to \lambda(B) = \lambda_0 + \tilde{\lambda}(B), \tag{25}$$

with $\tilde{b}$ and $\tilde{\lambda}$ the components of the order parameters that vary with changing field and vanish for $B = 0$. We assume these vanish continuously as the field is switched off, so we can consider a regime where $\left| \tilde{b} \right|$ and $\left| \tilde{\lambda} \right|$ are small compared to the zero-field parts.

## 4.1 Oscillations in the Linearized Theory

The free energy of the Kondo system at zero temperature, $\Omega_K$, is given by the sum over energies of the lower band, plus the final term in Eq. (20), giving an additional contribution from the mean fields. Because we must include spin when discussing a Kondo system, the band contribution to $\Omega_K$ is the same as Eq. (3) but with an additional sum over the spin degree of freedom, amounting to a factor of 2. As in the case of the excitonic insulator, the free energy has an implicit dependence on $B$ through the mean field functions $\tilde{b}(B)$ and $\tilde{\lambda}(B)$, so we expand around $(b, \lambda) = (b_0, \lambda_0)$ and keep terms up to first order in oscillatory quantities to find the first corrections on top of the band insulator result,

$$\Omega_K(B, b, \lambda) \approx \Omega_{KR} + \frac{\partial \Omega_{KR}}{\partial b_0} \tilde{b} + \frac{\partial \Omega_{KR}}{\partial \lambda_0} \tilde{\lambda} + \frac{1}{2} \frac{\partial^2 \Omega_{KR}}{\partial b_0^2} \tilde{b}^2 + \frac{1}{2} \frac{\partial^2 \Omega_{KR}}{\partial \lambda_0^2} \tilde{\lambda}^2 + \frac{1}{2} \frac{\partial^2 \Omega_{KR}}{\partial b_0 \partial \lambda_0} \tilde{b} \tilde{\lambda}$$

$$\approx \Omega_{KR0} + \tilde{\Omega}_{KR} + \frac{\partial \tilde{\Omega}_{KR}}{\partial b_0} \tilde{b} + \frac{\partial \tilde{\Omega}_{KR}}{\partial \lambda_0} \tilde{\lambda} + \frac{1}{2} \frac{\partial^2 \Omega_{KR0}}{\partial b_0^2} \tilde{b}^2 + \frac{1}{2} \frac{\partial^2 \Omega_{KR0}}{\partial \lambda_0^2} \tilde{\lambda}^2 + \frac{1}{2} \frac{\partial^2 \Omega_{KR0}}{\partial b_0 \partial \lambda_0} \tilde{b} \tilde{\lambda} \,. \tag{26}$$

We define $\Omega_{KR} = \Omega_K(B, b_0, \lambda_0)$ to be the free energy evaluated with rigid bands, which we then separate into constant $\Omega_{KR0}$ and oscillatory $\tilde{\Omega}_{KR}$ parts. The form of $\tilde{\Omega}_{KR}$ is identical to Eq. (5) up to an overall factor of 2 due to spin, the replacement $g \to b_0 V$, and $n^* = (\epsilon_d + \lambda_0 + 4\eta t b_0^2)/\omega^* - \gamma$ with $\omega^* = (1 + \eta b_0^2)\omega_c$. The vanishing of the first derivatives of $\Omega_{KR0}$ with respect to $b_0$ and $\lambda_0$ is synonymous with working at the level of mean field theory, and as a result we see that the contributions to the free energy from magnetic-field-induced oscillations of the mean field parameters enter at second order in small oscillations. We now seek to determine the size of these terms and their dependence on $B$, we did for the excitonic insulator.

We begin by examining $\tilde{b}(B)$ and $\tilde{\lambda}(B)$. We can evaluate the forms of these functions by analyzing the constraint equations, which for nonzero field have the same form as Eq. (21) and Eq. (22) but with the standard replacements we have made throughout,

$$n^f(B) = 1 - b(B)^2, \tag{27}$$

$$C(B) + \eta b(B)\left(K^f(B) + 4t\, n^f(B)\right) = -\lambda(B)b(B), \tag{28}$$

now with $n^f$, $C$, and $K^f$ functions of $B$ both explicitly and through their dependence on $b(B)$ and $\lambda(B)$. We now expand these functions around the rigid band case up to first order in small oscillatory quantities. For $n^f$ we have

$$n^f(B, b, \lambda) \approx n_0^f + \tilde{n}_R^f(B) + \frac{\partial n_0^f}{\partial b_0}\tilde{b}(B) + \frac{\partial n_0^f}{\partial \lambda_0}\tilde{\lambda}(B), \tag{29}$$

where $n_0^f$ is the $f$-electron density for $B = 0$, equal to the constant part of $n^f(B, b_0, \lambda_0)$, and we define $\tilde{n}_R^f(B)$ to be the oscillatory part of $n^f(B, b_0, \lambda_0)$. The same expansion can be done for $C(B)$ and $K^f(B)$, letting us similarly define the quantities $\tilde{C}_R(B)$ and $\tilde{K}_R^f(B)$.

We now expand Eqs. (27) and (28) up to first order in small oscillations. The terms at zeroth order are precisely Eqs. (21) and (22). We are left with the oscillatory components, obeying

$$\begin{pmatrix} \tilde{n}_R^f(B) \\ \tilde{C}_R(B) + \eta b_0 \tilde{K}_R^f(B) \end{pmatrix} = -\begin{pmatrix} u_b & u_\lambda \\ v_b & v_\lambda \end{pmatrix}\begin{pmatrix} \tilde{b}(B) \\ \tilde{\lambda}(B) \end{pmatrix}, \tag{30}$$

with

$$u_b = 2b_0 + \frac{\partial n_0^f}{\partial b_0}, \tag{31}$$

$$u_\lambda = \frac{\partial n_0^f}{\partial \lambda_0}, \tag{32}$$

$$v_b = \lambda_0 + \frac{\partial C_0}{\partial b_0} + \eta\left(K_0^f + b_0\frac{\partial K_0^f}{\partial b_0} + 4t(1 - 3b_0^2)\right), \tag{33}$$

$$v_\lambda = b_0 + \frac{\partial C_0}{\partial \lambda_0} + \eta b_0\frac{\partial K_0^f}{\partial \lambda_0}. \tag{34}$$

This system of equations can be inverted in general, and doing so gives $\tilde{b}$ and $\tilde{\lambda}$ as linear combinations of $\tilde{n}_R^f$ and $\tilde{C}_R + \eta b_0\tilde{K}_R^f$. The task is then to evaluate these quantities, for which we have explicit expressions.

Using Eqs. (21), (23) and (24) with the standard replacements for the case of $B \neq 0$, and evaluating all quantities at $(b, \lambda) = (b_0, \lambda_0)$, we arrive at expressions for $n^f(B, b_0, \lambda_0)$, $C(B, b_0, \lambda_0)$, and $K^f(B, b_0, \lambda_0)$, from which we can extract the function $\tilde{n}_R^f$, $\tilde{C}_R$, and $\tilde{K}_R^f$ using the Poisson summation formula.

Using the same methods we employed in Section 3 (see Appendix B), we find

$$
\begin{aligned}
\tilde{n}_R^f(B) &\approx -8b_0 V \frac{n_\Phi}{\omega^*} \sum_{p=1}^{\infty} \sin(2\pi p n^*) K_1\left(2\pi p \frac{2b_0 V}{\omega^*}\right) \\
&\sim -\sqrt{\frac{8b_0 V}{\omega^*}} n_\Phi \sum_{p=1}^{\infty} \frac{\sin(2\pi p n^*)}{\sqrt{p}} e^{-2\pi p \frac{2b_0 V}{\omega^*}},
\end{aligned}
\tag{35}
$$

for the oscillatory part of the $f$-electron density,

$$
\begin{aligned}
\tilde{C}_R(B) &\approx -8b_0 V^2 \frac{n_\Phi}{\omega^*} \sum_{p=1}^{\infty} \cos(2\pi p n^*) K_0\left(2\pi p \frac{2b_0 V}{\omega^*}\right) \\
&\sim -V\sqrt{\frac{8b_0 V}{\omega^*}} n_\Phi \sum_{p=1}^{\infty} \frac{\cos(2\pi p n^*)}{\sqrt{p}} e^{-2\pi p \frac{2b_0 V}{\omega^*}},
\end{aligned}
\tag{36}
$$

for the oscillatory part of the interband correlation, and

$$
\tilde{K}_R^f(B) \approx -\frac{\epsilon_d + \lambda_0 + 4\eta t b_0^2}{1 - \eta b_0^2} \tilde{n}_R^f(B),
\tag{37}
$$

for the oscillatory part of the $f$-electron kinetic energy. The asymptotic forms in second lines of Eqs. (35) and (36) apply in the regime where $\omega^* \ll 2b_0 V$. We see that, as has been true for all oscillatory quantities we have evaluated thus far, $\tilde{n}_R^f$, $\tilde{C}_R$, and $\tilde{K}_R^f$ all have a leading $B^{1/2}$ dependence and the $p^{\text{th}}$ harmonic is accompanied by $p$ powers of the Dingle factor. From Eq. (30), $\tilde{b}$ and $\tilde{\lambda}$ are linear combinations of these functions, so it follows that they share the same $B^{1/2}$ dependence and the same Dingle factor structure, which are also true of the derivative of $\tilde{\Omega}_{KR}$ appearing in Eq. (26).

Using these insights we can draw important conclusions about the additional oscillatory contribution of the free energy Eq. (26) without explicitly inverting Eq. (30), calculating the constants $u_b, u_\lambda, v_b$, and $v_\lambda$, or taking the second derivatives of $\Omega_{KR0}$. All of the oscillatory quantities comprising these additional terms go as $B^{1/2}$ and their lowest harmonics ($p = 1$) are proportional to a single power of the Dingle factor. The largest terms they contribute to in Eq. (26) are second order in oscillatory quantities, so they have a coefficient that is linear in $B$ and contribute at the same order as the $p = 2$ term of $\tilde{\Omega}_{KR}$, which has a coefficient going as $B^{3/2}$. Therefore, for weak fields these new terms are the dominant contribution to the second harmonic of the free energy, and the same sort of argument as at the end of Section 3 suggests that this is true for all higher harmonics as well. We also note that, also as for the excitonic insulator case, the behavior of these functions is determined by a single dimensionless parameter $\omega^*/b_0 V$, so that when these oscillations are present they provide a non-negligible contribution.

## 5 Discussion and Conclusion

We have shown that the field-induced oscillatory components of the mean field parameters in excitonic and Kondo insulator models yield qualitatively similar contributions to the oscillatory part of the free energy, and both systems differ from the band insulator in similar ways. In both cases oscillations of the mean field order parameters generate the dominant contributions to the second and higher QO harmonics for weak fields, which should have observable consequences in measurements of the de Haas-van Alphen effect. In particular, our results

demonstrate that measuring the field dependence of these higher harmonics allows one to distinguish between a simple band insulator and an insulating system with bands that are strongly affected by interactions. Additionally, since both the rigid band insulator and mean field contributions to the free energy, and therefore all thermodynamic quantities, are parametrized by the same dimensionless parameter, the oscillations of the self-consistent mean field parameters are always relevant when present and produce a distinct functional dependence on the magnetic field strength to second harmonic and higher oscillations.

Importantly, however, several features of the free energy are entirely insensitive to the mean field parameters acquiring weak magnetic field dependence. First, the lowest QO harmonic is unchanged from the behavior predicted by a rigid band model, which is guaranteed since the mean field state is defined as the saddle point of the free energy. Second, there are no changes to the nature of the Dingle factor–exponential sensitivity to the size of the $B = 0$ gap is the same as predicted from the theory of QO in a rigid band insulator. Thus, our results demonstrate that for interacting insulators the non-rigidity of the band structure with changing magnetic field strength does not preclude the use of the Dingle damping of QO as a means to measure properties of the gapped band structure at zero field.

Though we have focused here on the free energy, it is worth also considering other experimentally accessible quantities. First, the vanishing of mean field contributions to the fundamental frequency oscillation only applies to the free energy and thermodynamic quantities. In general, other quantities like the conductivity may have additional contributions at first order from the effects we study here. Second, there have been a number of works examining the specific heat and thermal transport measured in certain Kondo insulators, which are more akin to what would be expected in metals and have been attributed to neutral in-gap states such as excitons [18] or impurity bands [26], or neutral Fermi surfaces resulting from fractionalized electronic degrees of freedom [12, 13, 20, 22, 23, 28]. Our work here suggests that replacing rigid band structures with mean-field bands dependent on $B$ in those models that rely on band geometry may have qualitatively important effects.

Our results emphasize that QOs of the magnetization provide rich detail on the nature of the electronic state of band insulators. Measurements of the Dingle damping factor are particularly valuable. For materials that fall in the category of conventional band-insulators–including those where the band-gap includes self-consistent mean-field contributions–there should be agreement between the Dingle damping factor of the first harmonic and the electronic hybridization gap in the zero field limit. Disagreement would be an indication of the relevance of physics beyond what is captured by the mean-field models we have considered here.

# Acknowledgements

We thank Johannes Knolle for helpful discussions. We would also like to acknowledge Brian Skinner, Trithep Devakul, and Yves Hon Kwan for their insightful comments and questions.

**Data access** Data sharing not applicable to this article as no datasets were generated or analysed during the current study.

**Funding information** This work is supported by EPSRC Grant No. EP/P034616/1 and by a Simons Investigator Award.

## A  Comparison with Previous Results

Here we confirm that our $T = 0$ result for the free energy agrees with the $T \to 0$ limit of Eq.(8) in Ref. [11]. The system considered therein assumed an infinite valence band mass, corresponding to $\eta = 0$ here. In the notation used here, setting the chemical potential to lie inside the gap, and correcting for a missing factor of 2 and alternating sign in that equation, the oscillatory part of the free energy obtained there is

$$\tilde{\Omega}_R(T) = 2Tn_\Phi \sum_{p=1}^{\infty} \frac{(-1)^p}{p} \cos\left(2\pi p \frac{\epsilon_0}{\omega_c}\right) \sum_{n=0}^{\infty} \exp\left[-\frac{4\pi^2 pT}{\omega_c}\left(n+\frac{1}{2}\right) - \frac{pg^2}{\omega_c T}\frac{1}{n+\frac{1}{2}}\right]. \quad (38)$$

Define the dimensionless quantity $t_n = T(n+1/2)/\omega$, so that in the $T \to 0$ limit the sum over $n$ becomes an integral over $t$,

$$\tilde{\Omega}_R(T \to 0) \to 2\omega_c n_\Phi \sum_{p=1}^{\infty} \frac{(-1)^p}{p} \cos\left(2\pi p \frac{\epsilon_0}{\omega_c}\right) \int_0^\infty dt\, \exp[pf(t)], \quad (39)$$

where

$$f(t) = -4\pi^2 t - \frac{g^2}{\omega_c^2 t}. \quad (40)$$

One may recognize the resulting integral as being proportional the modified Bessel function of the second kind, $K_1(2\pi p g/\omega_c)g/(\pi\omega_c)$. Alternatively, the integral can be evaluated by the method of steepest descent to directly find the form for $g \gg \omega_c$. The saddle point is given by

$$f'(t^*) = 0 \Rightarrow t^* = \frac{g}{2\pi\omega_c}, \quad (41)$$

letting us approximate $f(t)$ as

$$f(t) \approx f(t^*) + \frac{1}{2}f''(t^*)(t - t^*)^2 \quad (42)$$

inside the integral, which is then of Gaussian form and can be evaluated to give

$$\tilde{\Omega}_R(T \to 0) = \sqrt{\frac{|g|\omega_c}{2}}\frac{n_\Phi}{\pi}\sum_{p=1}^{\infty}\frac{(-1)^p}{p^{3/2}}\cos\left(2\pi p\frac{\epsilon_0}{\omega_c}\right)e^{-2\pi\frac{2|g|}{\omega_c}}. \quad (43)$$

Recalling that $n^* = \epsilon_0/\omega^* - \gamma$ and $\omega^* = (1+\eta)\omega_c$, this exactly matches Eq. (5) for $\eta = 0$ and $\gamma = 1/2$.

## B  Evaluation of Oscillatory Functions

Functions written as a sum over Landau level indices can be divided into oscillatory and non-oscillatory parts using the Poisson summation formula. As a demonstration of the general procedure, here we provide the explicit calculation of the functions $\alpha_0$, $\tilde{\alpha}(B)$, and $\beta_0$, which then give $\tilde{\Delta}(B)$ as in Eq. (13).

Introduce the notation $E(n+\gamma) \equiv \epsilon_n^c - \epsilon_n^v = (n+\gamma)\omega^* - \epsilon_0 = (n-n^*)\omega^*$, with $n^* = \epsilon_0/\omega^* - \gamma$. Then we have

$$\alpha(B) = n_\Phi \sum_{n=0}^{\infty}\frac{1}{\sqrt{E(n+\gamma)^2 + 4\Delta_0^2}} = n_\Phi \int_0^\infty dx\,\frac{1}{\sqrt{E(x)^2 + 4\Delta_0^2}} + 2n_\Phi \int_0^\infty dx \sum_{p=1}^{\infty}\frac{\cos(2\pi p(x-\gamma))}{\sqrt{E(x)^2 + 4\Delta_0^2}}. \quad (44)$$

The first term in the second equality is what we call $\alpha_0$ and the second term is $\tilde{\alpha}(B)$. Putting $\omega^* x = k^2/2m_c$ in $\alpha_0$ we find

$$\alpha_0 = \int_0^\infty \frac{dk}{2\pi} k \frac{1}{\sqrt{\left(\epsilon_{\mathbf{k}}^c - \epsilon_{\mathbf{k}}^v\right)^2 + 4\Delta_0^2}}, \tag{45}$$

and we see that setting this equal to $1/V$ as in Eq. (12) is precisely equivalent to the $B = 0$ gap equation, Eq. (8), at least for the isotropic, parabolic dispersion implicitly assumed with this change of variables.

Now for $\tilde{\alpha}(B)$, the change of variables $z = x - \epsilon_0/\omega^* = x - n^* - \gamma$ gives

$$\tilde{\alpha}(B) = \frac{2n_\Phi}{\omega^*} \int_{-n^*-\gamma}^\infty dz \sum_{p=1}^\infty \frac{\cos\left(2\pi p(z + n^*)\right)}{\sqrt{z^2 + \left(\frac{2\Delta_0}{\omega^*}\right)^2}}. \tag{46}$$

We now assume that many Landau levels are occupied, i.e. $\epsilon_0 \gg \omega^*$, implying $n^* \gg 1$, which allows us to extend the lower limit of integration to $-\infty$. Rewriting the cosine as a sum of exponentials we then have

$$\tilde{\alpha}(B) \approx \frac{n_\Phi}{\omega^*} \sum_{p=1}^\infty e^{2\pi i p n^*} \int_{-\infty}^\infty dz \frac{e^{2\pi i p z}}{\sqrt{z^2 + \left(\frac{2\Delta_0}{\omega^*}\right)}} + \text{c.c.}, \tag{47}$$

where c.c. means the complex conjugate of the given term, and we see that the integral has become a Fourier transform which gives a modified Bessel function of the second kind. Combining the two terms we then arrive at

$$\tilde{\alpha}(B) \approx \frac{4n_\Phi}{\omega^*} \sum_{p=1}^\infty \cos(2\pi p n^*) K_0\left(2\pi p \frac{2\Delta_0}{\omega^*}\right). \tag{48}$$

We now need $\beta_0$, the non-oscillatory part of

$$\begin{aligned}
\beta(B) &= -n_\Phi \sum_{n=0}^\infty \frac{4\Delta_0}{\left(E(n+\gamma)^2 + 4\Delta_0^2\right)^{3/2}} \\
&= -n_\Phi \int_0^\infty dx \frac{4\Delta_0}{\left(E(x)^2 + 4\Delta_0^2\right)^{3/2}} - 2n_\Phi \int_0^\infty dx \sum_{p=1}^\infty \frac{4\Delta_0 \cos\left(2\pi p(x-\gamma)\right)}{\left(E(x)^2 + 4\Delta_0^2\right)^{3/2}},
\end{aligned} \tag{49}$$

which is the first term in the second line. Making the same changes of variables as above, then similarly extending the lower limit of integration to $-\infty$ we find

$$\beta_0 = -4\Delta_0 n_\Phi \int_{-\infty}^\infty dz \frac{1}{\left(z^2 + \left(\frac{2\Delta_0}{\omega^*}\right)^2\right)^{3/2}} = -\frac{2n_\Phi}{\Delta_0 \omega^*}. \tag{50}$$

Combining Eqs. (48) and (50) as in Eq. (13) we find precisely the form of $\tilde{\Delta}(B)$ in Eq. (14).

# C  Excitonic Insulator Temperature Dependence

Here we find the leading nonzero temperature corrections to the $T = 0$ results presented in the main text for the excitonic insulator. At nonzero $T$ the mean field free energy is

$$\Omega_X = \frac{\Delta^2}{V} - T \int_{-\infty}^{\infty} d\epsilon \, g(\epsilon) \ln\left(1 + e^{-\epsilon/T}\right), \tag{51}$$

$$g(\epsilon) = \frac{1}{N} \sum_{\mathbf{k},\alpha} \mathcal{A}(\epsilon - E_\alpha(\mathbf{k})), \tag{52}$$

where $g(\epsilon)$ is the density of states, written in terms of the spectral density $\mathcal{A}$ which is simply a $\delta$-function in the absence of disorder. Including a nonzero magnetic field via the prescriptions already discussed and employing the Poisson summation formula we find

$$g(\epsilon) = \frac{2n_\Phi}{\omega^*} \sqrt{\frac{\epsilon^2}{\epsilon^2 - \Delta^2}} \Theta(\epsilon^2 - \Delta^2) \left\{ 1 + 2 \sum_{p=1}^{\infty} \cos\left[ 2\pi p \left( \frac{2\sqrt{\epsilon^2 - \Delta^2} + \epsilon_0}{\omega^*} + \gamma \right) \right] \right\}, \tag{53}$$

where $\Delta = \Delta(B, T)$ is the full field- and temperature-dependent gap function, and $\Theta(x)$ is the Heaviside theta function, which equals 1 for $x > 0$ and vanishes otherwise. Here the theta function gives the gap in the spectrum–there are no states for energies with $|\epsilon| < |\Delta|$. Using this form of the density of states in Eq. (51) and changing to a new integration variable $\xi$ defined through $\epsilon = \sqrt{\xi^2 + \Delta^2}$ we obtain

$$\begin{aligned}
\Omega(B, T) = \frac{\Delta^2}{V} - \frac{2n_\Phi T}{\omega^*} \int_0^\infty d\xi \ln\left[ 2\left( 1 + \cosh\left( \frac{\sqrt{\xi^2 + \Delta^2}}{T} \right) \right) \right] \\
\times \left\{ 1 + 2 \sum_{p=1}^\infty \cos\left[ 2\pi p \left( \frac{2\xi + \epsilon_0}{\omega^*} + \gamma \right) \right] \right\}.
\end{aligned} \tag{54}$$

As noted above, the gap $\Delta$ is itself a function of temperature, and for $B = 0$ obeys

$$\frac{1}{V} = \nu \int_0^\infty d\xi \frac{\tanh\left( \frac{\sqrt{\xi^2 + \Delta^2}}{2T} \right)}{2\sqrt{\xi^2 + \Delta^2}}, \tag{55}$$

where $\nu$ is the density of states for free electrons in two dimensions. We can expand around $T = 0$ to give

$$\tanh\left( \frac{\sqrt{\xi^2 + \Delta^2}}{2T} \right) \approx 1 - 2e^{-\sqrt{\xi^2 + \Delta^2}/T}, \tag{56}$$

allowing us to separate the gap equation into a temperature independent ($T = 0$) part, determining the zero-temperature value of the gap, $\Delta(T = 0)$, and a nonzero temperature part providing a correction to $\Delta$ that is exponentially small for temperatures $T \ll \Delta(T = 0)$. (The same can be done for $B \neq 0$ as well.) This defines what we mean by the low-temperature regime.

Returning now to the free energy, we can approximate the temperature dependent factor in the low temperature regime,

$$\begin{aligned}
T \ln\left[ 2\left( 1 + \cosh\left( \frac{\sqrt{\xi^2 + \Delta^2}}{T} \right) \right) \right] = \sqrt{\xi^2 + \Delta^2} + 2T \ln\left( 1 + e^{-\sqrt{\xi^2 + \Delta^2}/T} \right) \\
\approx \sqrt{\xi^2 + \Delta^2} - 2T e^{-\sqrt{\xi^2 + \Delta^2}/T}.
\end{aligned} \tag{57}$$

With this we then separate Eq. (54) into two terms. It is straightforward to confirm that the $T$-independent term reproduces what is found for the $T = 0$ free energy of the excitonic insulator after applying the Poisson summation formula. The second term then contains the entirety of thermally activated contribution to the free energy, which we see is exponentially suppressed–the largest this term can be is $T \exp(-\Delta(T = 0)/T) \ll 1$. Thus, in the low temperature regime $T \ll \Delta(T = 0)$ the zero-temperature calculations we have provided in the main text are accurate up to exponentially small corrections.

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
