# Peer review of "Quantum oscillations in interaction-driven insulators"

_SciPost Physics, doi:SciPost Phys. 12, 123 (2022)_

## Round 1 · Referee Report · Brian Skinner (Referee 1) · 2022-2-20

Report

This paper considers a timely and interesting question: whether quantum oscillations (QOs) of magnetization can appear in excitonic or Kondo insulators. It has recently been established that band insulators can exhibit QOs if the band structure has a "Mexican hat" shape or an otherwise sharp kink near the band bottom, such that the band edge itself forms a "surface" in momentum space. Here the authors demonstrate that when such band structures appear for purely interaction-driven reasons (as in excitonic insulators or Kondo insulators) the same type of QOs persist. Specifically, the lowest harmonic of QO is exactly the same as it would have been if the system were a rigid band insulator, even though the effective band parameters are magnetic-field-dependent. Interestingly, however, the higher harmonics are dominated by the magnetic-field-induced modulation of band parameters. These two results are both quite useful for the continued study of various seemingly-mysterious materials that exhibit quantum oscillations in an insulating state.

The paper is clearly written and it proceeds by examining three relatively simple but generic models. I think the results are generally clear and unimpeachable, and the limitations and assumptions are clearly stated.

I have only a small number of relatively straightforward and optional suggestions for the authors to consider:
1. It might be worth saying explicitly somewhere that you have set $\hbar = 1$.
2. I would have liked to see more discussion of how your results compare/contrast with those of Ref. 21 by Patrick Lee. Similar conclusions seem to have been reached there for the case of excitonic insulators, but I can't immediately tell whether there is any disagreement.
3. In the vicinity of equation (6) the authors say that they are considering a "two-dimensional model", but it seems like the results are generally applicable in 3D and I can't immediately see why the two-dimensionality is necessary.
4. The temperature dependence of the QO amplitude is discussed in Appendix C, but I couldn't see any discussion (or reference to this appendix) in the main text. Since the functional form of the temperature dependence of QOs is a major point of discussion for certain experiments (used to supposedly rule out or rule in certain explanations), it might be good for the authors to add a more prominent discussion to the main text. I think that the T-dependence here is not of the usual Lifshitz-Kosevich type, and elaborating on this point could be helpful.

  • validity: high
  • significance: high
  • originality: good
  • clarity: high
  • formatting: perfect
  • grammar: perfect

Author:  Andrew Allocca  on 2022-03-10  [id 2278]

(in reply to Report 1 by Brian Skinner on 2022-02-20)
Category:
answer to question

Thank you for this positive assessment and for these helpful comments. We answer these in the following, and have updated the paper accordingly.

1. It might be worth saying explicitly somewhere that you have set $\hbar$=1.

Thank you for pointing this out. Now fixed.

2. I would have liked to see more discussion of how your results compare/contrast with those of Ref. 21 by Patrick Lee. Similar conclusions seem to have been reached there for the case of excitonic insulators, but I can't immediately tell whether there is any disagreement.

There are several differences between our work and that of Patrick Lee in Ref. 21, but no obvious disagreement from what we can discern. The first difference is simply the quantities that we each consider; we evaluate the free energy of the system allowing insight into thermodynamic quantities, while he is focused on transport, specifically the conductivity of the thermally activated population of charge carriers, being more directly interested in the measurements of Wang et al. on WTe2. Second, the oscillation of the gap with field strength that he considers is quite different from what we calculate. He considers the oscillations of the gap that arise from Landau quantization of a “rigid” band structure of hybridized particle and hole bands–rigid in the sense that the hybridization is fixed to equal the excitonic condensate order parameter at B=0. He defines the gap for B>0 as the energy between the highest Landau level in the lower band and the lowest Landau level in the upper band. This is similar to the oscillations found in the framework of Ref. 10.

In the present paper, we go beyond Ref. 21 (and Ref. 10) by considering the oscillatory nature of the excitonic order parameter itself by solving the linearized gap equation in the presence of a magnetic field, which then causes the band structure to change with B. One of our results is that considering only the B=0 rigid band structure is indeed sufficient to analyze the fundamental oscillation frequency of the free energy and therefore of thermodynamic properties. However that argument does not apply more broadly. It seems generically true that non-thermodynamic quantities such as transport should have a first-order contribution from the effects we identify. That is, there is, in general, an additional oscillatory contribution to the activation gap used Ref. 21 coming from this self-consistency (see Eqn (14) of the present paper).

3. In the vicinity of equation (6) the authors say that they are considering a "two-dimensional model", but it seems like the results are generally applicable in 3D and I can't immediately see why the two-dimensionality is necessary.

We call our model two-dimensional because we do not include any dispersion along $k_z$, which would be present in a full three-dimensional model. However, there is a straightforward and well understood procedure to extend 2d results like those we present to a full 3d system (see e.g. Shoenberg, Ref. 23) and we expect that doing so would not change the fundamental nature of our results—the fundamental frequency will remain unaffected, and interactions will still dominate higher harmonics for weak fields.

4. The temperature dependence of the QO amplitude is discussed in Appendix C, but I couldn't see any discussion (or reference to this appendix) in the main text. Since the functional form of the temperature dependence of QOs is a major point of discussion for certain experiments (used to supposedly rule out or rule in certain explanations), it might be good for the authors to add a more prominent discussion to the main text. I think that the T-dependence here is not of the usual Lifshitz-Kosevich type, and elaborating on this point could be helpful.

The T-dependence we find for the case of the excitonic insulator is certainly not of the usual Lifshitz-Kosevich type–we find activated dependence for $T \ll \Delta_0$. However, the range of temperatures for which our analysis applies is limited to be much smaller than the gap (it is not immediately clear what the corresponding temperature range is for the case of a Kondo insulator because there are two mean field parameters, though $T\ll\Delta$ seems a good estimate). To compare to the full T-dependence in experiments we would need to push to higher temperatures, where it would be crucial to include the thermal occupation of the upper band and hence the full dependence of the gap on T. We have no analytical result for this, so the role of these interaction effects on temperature would therefore require significant numerical work to establish. Because we know the size of the gap decreases with temperature, therefore lessening the Dingle damping, it is hard to even say if the size of the effect would be enhanced or further suppressed, and it is likely that pushing to higher T would necessitate a simultaneous push to stronger fields, i.e. $\omega \sim \Delta$. This is certainly an interesting avenue of research to consider, but saying anything firm that could relate to experiment is beyond the scope of this work.

---

## Round 1 · Referee Report · Trithep Devakul (Referee 2) · 2022-2-21

Report

The quantum oscillation (QO) of thermodynamic properties as a function of an applied magnetic field, a phenomenon typically associated with metals, but has also been observed in insulating systems. Insulating systems displaying QOs are “proximate” to a metallic system with electron and hole Fermi surfaces that have been gapped out. In this manuscript, Allocca and Cooper examine the form of QOs in such insulators where the gap opening mechanism is of various origins. First, they examine the case where the gap is due to single-particle hybridization, and obtain an expression for the oscillations of the free energy as a function of magnetic field. They then consider two cases where the gapping mechanism is interaction-driven: an excitonic insulator and a Kondo insulator, which are treated in a mean-field approximation. Additional corrections arise in the QOs of these insulators due to the magnetic field dependence of the mean-field order parameters themselves. These corrections only appear for the second (and higher) harmonics of the QOs.

This paper provides a systematic examination of QOs in interaction-induced insulators, which is a valuable and timely contribution to the literature. The manuscript is excellent as is, but I just have a couple questions that the authors may want to consider.

1. The authors only examine the QOs in the free energy Omega. Since the mean-field order parameters are obtained (by definition) as stationary points of Omega, the oscillation of the mean-field order parameters only contribute to second order and higher. Quantities which derive directly from the free energy, such as magnetization, will therefore also only get 2nd and higher order contributions. Can the authors comment on whether other quantities (such as conductivity in the Shubnikov-de Haas effect) which are not simple functions of the free energy, may show QOs with interaction effects potentially visible in first order?

2. The QOs in such insulators are exponentially damped at small fields (Eq 5), but the interaction effects are also dominant in the 2nd harmonic at weak fields (after Eq 16). Is it possible that the QOs are exponentially suppressed in the regime where interaction effects are dominant, making them experimentally inaccessible? The experimental relevance of interaction-induced effect will be greatly clarified if the authors can give a rough order of magnitude of the various contributions to the QOs in a realistic system (e.g. SmB6) at various magnetic fields.

  • validity: high
  • significance: high
  • originality: high
  • clarity: high
  • formatting: excellent
  • grammar: excellent

Author:  Andrew Allocca  on 2022-03-10  [id 2279]

(in reply to Report 2 by Trithep Devakul on 2022-02-21)
Category:
answer to question

Thank you for this positive assessment and for these helpful comments. See detailed answers below. We have also updated the paper to reflect these comments.

1. The authors only examine the QOs in the free energy Omega. Since the mean-field order parameters are obtained (by definition) as stationary points of Omega, the oscillation of the mean-field order parameters only contribute to second order and higher. Quantities which derive directly from the free energy, such as magnetization, will therefore also only get 2nd and higher order contributions. Can the authors comment on whether other quantities (such as conductivity in the Shubnikov-de Haas effect) which are not simple functions of the free energy, may show QOs with interaction effects potentially visible in first order?

It is a very real possibility that non-thermodynamic quantities such as conductivity could have a nonzero contribution at first order from the field dependence of the mean field parameters. Unlike for the free energy there is no obvious argument why such a contribution should vanish in general, and any quantity that depends on the mean field parameters may therefore acquire an additional oscillatory component at first order. The nature of such a contribution would of course need to be analyzed on a case-by-case basis.

2. The QOs in such insulators are exponentially damped at small fields (Eq 5), but the interaction effects are also dominant in the 2nd harmonic at weak fields (after Eq 16). Is it possible that the QOs are exponentially suppressed in the regime where interaction effects are dominant, making them experimentally inaccessible? The experimental relevance of interaction-induced effect will be greatly clarified if the authors can give a rough order of magnitude of the various contributions to the QOs in a realistic system (e.g. SmB6) at various magnetic fields.

Our prediction is about the general functional form of the free energy (or magnetization), with only one dimensionless parameter: the quantity $B/B_0$ that enters the Dingle damping (with magnetic field $B$ and the material-dependent field-scale $B_0$) is the same parameter that controls the size of the interaction corrections. Thus, if the magnetic field is sufficiently large (i.e. $B \sim B_0$) such that the second harmonic can be measured at all in the weak field regime, which is a nontrivial experimental task, one expects also a significant deviation from the rigid band theory arising from the interaction correction we have derived. Even keeping additional terms in the weak field expansion of the Bessel functions, which gives the exponential form of the Dingle factor, the magnetization with and without these interaction effects have distinct functional forms.

---

## Round 1 · Referee Report · Yves Hon Kwan (Referee 3) · 2022-2-23

Report

The intriguing appearance of quantum oscillations (QOs) in insulators has been studied experimentally and theoretically within the last decade. Many of these theoretical proposals proceed by analyzing the impact of the magnetic field (B) on an effective "rigid" band model. However for several systems of interest, the underlying zero-field state owes its insulating character to interaction-induced correlation effects, which may themselves depend on B. In the present work, the authors address this for two simple models of the Kondo and excitonic insulators. Within a mean-field treatment at T=0, they extract the leading oscillatory parts of the free energy in the weak oscillation regime, accounting for the B-dependence of the mean-field parameters. Their main finding is that for both models, the lowest harmonic as well as the exponential "Dingle" factor for all harmonics remains unchanged, but corrections with a stronger (for small fields) overall power of B are generated for the higher harmonics.

The topic of the paper is relevant, especially given recent experimental results in both Kondo and excitonic materials. I believe that the authors have made an important step in understanding the possible behaviours of QOs in such platforms. The analytical results look solid to me, and the methods and approximations are clear.

Some comments and questions for the authors to consider:

1. The conclusions here seem applicable to magnetization and heat capacity. Can the analysis here be applied to oscillations in transport? I think it would be useful to comment on this somewhere, given the current mysteries surrounding resistivity oscillations in materials like WTe2.

2. It is emphasized that the exponential Dingle factor is unchanged, and hence it can still be used to extract the zero-field gap. To get the absolute value of the gap, it looks like one would still need to obtain an independent measure of the effective masses.

3. Is it obvious why there isn't any hybridization between conduction and valence levels of different Landau level indices?

4. As far as I can tell, Appendix C is not referenced in the main text. Could one make a statement here about deviation of the T-dependence (which has been observed in some experiments) from the usual aT/sinh(aT) form expected for metals?

5. If it is possible for one of the materials, it would be nice to plug in numbers and give a rough estimate of the magnetic field scale below which the analysis holds. This shouldn't be too small otherwise the oscillations would be too weak to resolve (and it looks like one needs enough oscillation periods to extract the power of B on top of the exponential suppression).

6. In the final section, it may be worth discussing potential avenues for future study based on this work, e.g. could fluctuations beyond mean-field be captured in this formalism?

  • validity: high
  • significance: high
  • originality: good
  • clarity: top
  • formatting: excellent
  • grammar: excellent

Author:  Andrew Allocca  on 2022-03-10  [id 2280]

(in reply to Report 3 by Yves Hon Kwan on 2022-02-23)
Category:
answer to question

Thank you for this positive assessment and for these helpful comments. We address these in detail in the following, and have updated the paper accordingly.

1. The conclusions here seem applicable to magnetization and heat capacity. Can the analysis here be applied to oscillations in transport? I think it would be useful to comment on this somewhere, given the current mysteries surrounding resistivity oscillations in materials like WTe2.

See our response to question 1 in Report 2 by Trithep Devakul.

2. It is emphasized that the exponential Dingle factor is unchanged, and hence it can still be used to extract the zero-field gap. To get the absolute value of the gap, it looks like one would still need to obtain an independent measure of the effective masses.

What experiments could extract from field measurements of the Dingle damping factor is indeed a parameter $B_0$ that is proportional to the hybridization gap times the cyclotron mass. Similarly, the oscillation frequency can be viewed as proportional to the product of the cyclotron mass and another characteristic energy scale of the band structure–in our models, the energy offset between the band edges at zero momentum. These two quantities ($B_0$ and the frequency) characterize the band structure of the system, and knowing either energy scale or the effective mass would then give values for the others, though the ratio of these two parameters could be extracted without additional information.

3. Is it obvious why there isn't any hybridization between conduction and valence levels of different Landau level indices?

The lack of hybridization between different Landau level indices follows from the assumption of a spatially homogeneous mean field state producing spatially uniform hybridization between the bands.

4. As far as I can tell, Appendix C is not referenced in the main text. Could one make a statement here about deviation of the T-dependence (which has been observed in some experiments) from the usual aT/sinh(aT) form expected for metals?

See our response to question 4 of Report 1 by Brian Skinner.

5. If it is possible for one of the materials, it would be nice to plug in numbers and give a rough estimate of the magnetic field scale below which the analysis holds. This shouldn't be too small otherwise the oscillations would be too weak to resolve (and it looks like one needs enough oscillation periods to extract the power of B on top of the exponential suppression).

See our response to question 2 of Report 2 by Trithep Devakul.

6. In the final section, it may be worth discussing potential avenues for future study based on this work, e.g. could fluctuations beyond mean-field be captured in this formalism?

Thank you, this is also a good suggestion. In light of the number of mechanisms proposed to explain various other aspects of experiments on Kondo insulators, one obvious avenue for future study is to consider how the effects of non-rigid band structures propagate forward through those theories. The role of fluctuations of the mean-field order is also something worthwhile to consider.

---

## Round 2 · Author Response

We would like to thank the referees for their careful reading of our paper, their insightful comments and questions, and positive assessment of the work we present. In the resubmission we have made additions and revisions, outlined below, to address the points they raised in their reports, and hope that after doing so the paper is acceptable for publication in SciPost Physics.

Thank you,
Andrew Allocca and Nigel Cooper

---

## Round 2 · List of Changes

• Added several additional references to relevant work in the introduction
  • Added a brief explanation in Section 2 of why we say we are working in 2d, and how our work could be extended to 3d
  • Added a comment and references about how the mean field formalism we use can be extended to consider fluctuations
  • Added a discussion in Section 3 of how our work relates to that in Ref. 21 (Refs. 28 and 29 in this version)
  • Added an explanation in Section 3 of how our results depend only on a single dimensionless quantity and what that means for the observability of the effects we find, regardless of material parameters
  • Added subsection 3.2 discussing the effects of nonzero temperature beyond what we explicitly examine in Appendix C
  • Added a paragraph in the conclusion about the effect our results may have for non-thermodynamic quantities
  • Fixed minor typos and wording throughout

---

## Editorial Decision

published